# Is Radiofrequency Ablation Superior to Intra-Articular Injections for the Treatment of Symptomatic Knee Osteoarthritis?—A Systematic Review

**DOI:** 10.3390/jpm13081227

**Published:** 2023-08-03

**Authors:** Byron Chalidis, Pericles Papadopoulos, Panagiotis Givissis, Charalampos Pitsilos

**Affiliations:** 11st Orthopaedic Department, Aristotle University of Thessaloniki, 57010 Thessaloniki, Greece; pgivissis@gmail.com; 22nd Orthopaedic Department, Aristotle University of Thessaloniki, 54635 Thessaloniki, Greece; perpap@otenet.gr (P.P.); xnostos@hotmail.com (C.P.)

**Keywords:** radiofrequency ablation, nerve, knee, osteoarthritis, pain, injection, knee

## Abstract

The radiofrequency ablation (RFA) is considered a valid, minimally invasive treatment modality for the management of symptomatic knee osteoarthritis (OA). The aim of this study was to compare the outcomes of RFA with that of commonly used intra-articular injections for the persistent knee pain due to OA. Medline/Pubmed and Scopus databases were systematically searched up to April 2023 to identify studies comparing the effect of RFA and intra-articular injections (IAIs) on knee OA. Nine studies including 899 patients fulfilled the eligibility criteria and were included in the systematic review. The RFA procedure was related with improved knee pain relief compared to IAIs at 3-, 6- and 12-month follow-up (*p* < 0.001). Similarly, functional improvement was greater in RFA treatment than that observed after hyaluronic acid (HA), steroid or platelet-rich plasma (PRP) injections (Visual Analogue Scale *p* < 0.001, Numeric Rating Scale *p* = 0.019, Western Ontario and McMaster University Osteoarthritis Index *p* = 0.012). The overall procedural complication rate of RFA was 10.2% and was higher than steroid (*p* = 0.023) and PRP (*p* = 0.017) injections. However, no severe adverse events were reported. For patients with symptomatic knee OA, RFA seems to be more effective than IAIs in alleviating pain and improving joint function, despite the relatively higher incidence of non-serious adverse events. However, due to the limited number of studies and patients, this result should be interpreted with caution and not be generalized to the entire knee OA population.

## 1. Introduction

Osteoarthritis (OA) of the knee joint is a common cause of pain and disability in adults and older individuals [1]. For the early stages of the disease, different conservative treatment options have been introduced with variable success in terms of alleviating pain and improving joint function [2]. Activities modification, weight-loss, physiotherapy and muscle-strengthening exercises along with medication including non-steroid anti-inflammatory drugs, painkillers and oral opioids may reduce joint inflammation and pain and delay further surgical intervention with a total knee arthroplasty [3]. In this direction, intra-articular injections (IAIs) with cortisone, viscosupplementation or platelet-rich plasma (PRP) can provide short-term pain relief and can be administrated in either single or multiple doses [4]. However, their effectiveness and superiority of one treatment over another has not been clearly defined, and robust clinical evidence is still lacking [5].

Radiofrequency ablation (RFA) has been widely applied for the treatment of neuropathic pain, trigeminal neuralgia, cancer and spinal pain as it can reduce the conduction of peripheral pain stimulation to the central nervous system [6,7,8]. The RFA includes three different techniques: pulsed-RFA, thermal-RFA and cooled-RFA [9,10]. In the last decade, its use has also been expanded to patients suffering from osteoarthritic knee pain and particularly for those who are not suitable for total knee arthroplasty due to coexisting severe medical comorbidities or they are undesiring of surgery [11]. The technique is a low-risk and safe procedure that includes targeted thermal damage of nerve structures innervating painful tissues and interruption of the transmission of pain signals [12]. An electrode is placed on the target nerve and the applied thermo-coagulation induces tissue destruction by producing heat up to 60–80 °C [13,14]. The elevation of temperature leads to the separation of myelinated axons and reversibly impairs nerve function [15]. Although nerve regeneration is expected to occur after 9–12 months, patients may experience pain relief up to 2 years [16]. Even after that time period, the intensity of pain may be lower than that before the procedure [17].

Regarding the knee joint, RFA aims at the genicular branches of femoral, saphenous, common peroneal, obturator and tibial nerves that travel along the periosteal areas connecting the femoral shaft to bilateral epicondyles and the shaft of the tibia to medial epicondyle [18]. However, the durability and long-term efficacy of the technique is unpredictable and not well documented as pain is believed to be attenuated while the nerve structure is gradually restored [19,20]. Therefore, repeated ablations may be required in order to improve the magnitude and duration of treatment effect [20,21]. Previously, the effect of the RFA on function and knee osteoarthritis pain has been compared with other invasive or non-invasive treatment modalities without definitive conclusions [22]. Moreover, no systematic reviews or meta-analysis have been performed so far to directly compare the RFA with the various types of IAIs.

The aim of the current review was to present all the current evidence regarding the outcome of radiofrequency neurotomy in patients suffering from knee OA and identify any differences compared to IAI therapy, including viscosupplementation, corticosteroid and/or PRP.

## 2. Materials and Methods

### 2.1. Study Type

A systematic review was conducted under the Preferred Reporting Items for Systematic Review and Meta-Analysis (PRISMA) guidelines [23] and was registered in PROSPERO with the registration number CRD42023446442.

### 2.2. Search Strategy

Two electronic databases were used to identify potential studies and develop the systematic review: Medline/Pubmed and Scopus. These databases were searched for papers published until April 2023. Following “all-field” screening, the keywords that were utilized in the process of article search and retrieval included: “knee” AND “arthritis” OR “osteoarthritis” AND “radiofrequency” AND “ablation”.

### 2.3. Eligibility Criteria and Study Selection

Studies concerning the treatment of knee OA with RFA were eligible for further analysis. The inclusion criteria comprised of clinical studies comparing the efficacy of RFA with that of a single IAI in knee OA that were published in English language journals and contained at least five patients in each outcome category. Case reports, editorials, expert opinions and non-human studies were excluded from further analysis. The EndNote X9 software (Clarivate Analytics, Philadelphia, PA, USA) was used to remove the duplicate studies from the two databases. Moreover, the references of the initial search were also independently searched for possible missing articles. Two authors (C.P. and B.C.) investigated article’s titles and abstracts and assessed the eligibility. Disagreements and conflicts concerning inclusion were resolved by a third author (P.P.).

### 2.4. Data Extraction and Quality Assessment

Two authors (C.P. and B.C.) extracted the following data and information from each eligible article using Microsoft Excel database: first author’s last name; year of publication; number of patients; number of cases; gender; age; body mass index (BMI); duration of pain; grade of osteoarthritis according to Kellgren–Lawrence (K–L) imaging rating scale (Grade 1: no OA, Grade 2: mild OA, Grade 3: moderate OA, Grade 4: severe OA) [24]; follow-up; and functional parameters’ complications or adverse events. The primary outcome of the study was the comparison of the effect of RFA and IAIs on knee pain, using the Visual Analogue Scale (VAS) and the Numeric Rating Scale (NRS) scores.

The quality of the included trials was assessed using the Critical Appraisal Skills Programme (CASP) for the randomized control trials and the revised methodological index for non-randomized studies (MINORS) for the comparative studies [25]. The CASP includes 11 questions about the basic study design, the methodology, the results and the value of the results, with three possible answers for each question: “Yes” (2 points), “Can’t tell” (1 point) or “No” (0 points). The revised MINORS includes 12 questions with three possible answers for each on: “reported and adequate” (2 points), “reported but inadequate” (1 point) or “not reported” (0 points).

The risk of bias of each study was assessed using the Cochrane RoB 2 for randomized control trials and the ROBINS-I tool (2026) for the rests studies [26,27]. The risk of bias in the former can be characterized as “low”, “some concerns” or “high”, while in the latter as “low”, “moderate” or “serious”, “critical” or “no information”.

The certainty in the body of evidence was evaluated using the GRADE system [28]. This involves consideration of within-trial risk of bias (methodological quality), directness of evidence, heterogeneity, precision of effect estimates and risk of publication bias.

### 2.5. Data Synthesis and Analysis

All the extracted data were transcribed into SPSS (IBM Corp. Released 2017. IBM SPSS Statistics for Windows, Version 25.0. Armonk, NY, USA: IBM Corp.) and subsequently analyzed. Continuous variables were described either as means with standard deviations or as medians. Categorical variables were demonstrated as proportions. Sub-group column proportions were compared for categorical variables using z-test, and Bonferroni correction was applied to adjust for *p* values. All tests were two-sided and statistical significance was assumed at a *p* value of <0.05. Heterogeneity of between-study variance was assessed using Cochrane’s Q test (*p* < 0.10) and quantified using the *I*^2^ statistics. *I*^2^ > 50% and a *p* < 0.05 were considered substantial heterogeneity [29].

## 3. Results

### 3.1. Study Selection

We initially found 917 articles from two bibliographic databases based on the search strategy. After screening for duplicates, 798 articles were eliminated for irrelevant content, study design and lack of data. Further 763 articles were excluded after screening the titles and abstracts according to the exclusion criteria, and 35 full-text articles were retrieved and assessed for eligibility. Twenty-six articles were excluded due to insufficient data; fourteen were review articles, seven were animal studies, three were case reports and two were expert opinions. Finally, the selection process yielded nine articles eligible for inclusion in the systematic review (Figure 1).

Among the nine studies, seven were randomized control trials, one was prospective and another one was retrospective comparative study. According to the CASP, the mean score of the randomized control trials was 19.3 (range: 16–21) [13,14,30,31,32,33,34]. The MINORS score was 21 for the prospective and 20 for the retrospective comparative study [35,36].

### 3.2. Study Characteristics

Our literature search identified nine studies comparing the effect of RFA with different types of IAI, published in the last 10 years. The risks of bias of the included studies are summarized in Table 1. There was absence of heterogeneity as indicated by *I*^2^ (0%; *p* = 0.82). The mean duration of symptoms was 72.4 months (range: 32.5–128 months), and the mean follow-up period was 7 months (range: 3–12 months). The total number of patients treated with RFA was 452 (66.9% females) with a mean age of 59.6 years (range: 41–78) and a mean BMI of 27.4 kg/m^2^ (range: 19–37). The temperature of RFA ranged from 42 °C to 80 °C, the duration of application from 90 to 180 s and the cycles of RFA from 1 to 3.

There were 447 patients (68.2% females) treated with IAI. The mean age was 58.5 years (range: 38–76), and the mean BMI was 26.7 kg/m^2^ (range: 20–36). Of the 447 patients, 146 (32.7%) underwent a hyaluronic acid (HA) injection, 138 (30.9%) a corticosteroid injection, 100 (22.4%) a PRP injection, 36 (8.1%) an ozone injection and 27 (6%) had combined HA and PRP injections (Table 2).

The effect of RFA or IAI on patients with knee OA was evaluated using the following functional scores: the VAS; the NRS; the Western Ontario and McMaster University Osteoarthritis Index (WOMAC); the Global Perceived Effect (GPE); the American Knee Society Score (AKSS); the Index of Severity for Osteoarthritis of the Knee (IKS); the Lysholm Knee score (LKS); the Oxford Knee Scale (OKS); 36-Item Short Form Survey (SF-36); and EuroQol 5 Dimension 5 Level (EQ-5D-5L).

### 3.3. Outcomes

#### 3.3.1. RFA and Knee OA

The VAS was evaluated in six RFA studies including 261 patients out of the total 452 patients [13,14,32,33,34,36]. The baseline mean score was 7.7. This was improved to 3 after 3 months (5 studies—255 patients [14,32,33,34,36]), to 3.7 after 6 months (2/6 studies—149 patients [32,34]) and to 3.64 after 12 months (3 studies—185 patients) (*p* < 0.05) [13,32,34].

The NRS was measured in two studies, including 165 patients [30,31]. This was improved from 7.1 to 2.55 and 2.6 at 3 and 6 months, respectively (*p* < 0.05).

The WOMAC score was obtained in three studies including 138 patients [14,30,33]. The 3-month mean score was 28, which was significantly lower compared to the baseline 67.9 (*p* < 0.05).

The LKS was used in one study including 49 cases [34]. The baseline 42 points were improved to 85 points at 12-month follow-up (*p* < 0.05).

The GPE was evaluated in three studies with a total number of 191 patients [30,31,35]. After RFA of the painful knee, there was significant improvement in 79.5% and 81.5% of cases after 3 months and 6 months, respectively (*p* < 0.05).

Two different versions of the OKS were used in two studies. In one version, the improved outcome was associated with increased score, while in the other, the decreased value was related to better function. In the first study with 36 patients, the baseline score was improved from 41.2 to 22.6 at 12 months (*p* < 0.05) [13]. In the second study, including 76 patients, the score was improved from 16.7 to 35.7 between the baseline and the 6-month follow-up (*p* < 0.05) [31].

The EQ-5D-5L was used in one study, including 89 patients [30]. The baseline score was 0.67, and it was increased to 0.83 and 0.8 at 3 and 6 months, respectively (*p* < 0.05).

The SF-36 was evaluated in one study with 27 patients [36]. From 360 at baseline, it was increased to 434 at 3 months (*p* < 0.05).

All aspects of the AKSS were improved from baseline to 3-month follow-up in one study, which contained 27 patients (*p* < 0.05) [36]. Similarly, in another study with 100 patients, the IKS was improved from the baseline to the 12-month follow-up (*p* < 0.05) [32]. (Table 3).

#### 3.3.2. RFA vs. IAIS

In total, 452 patients were included in the RFA group and 447 patients in the IAI group. Regarding the VAS score, the improvement after RFA was significantly greater compared to IAI after 3, 6 and 12 months (*p* < 0.001). Following RFA, the VAS was reduced from a baseline of 7.7 to 3 at 3 months, to 3.7 at 6 months and to 3.6 at 12 months. In the IAI group, the VAS was decreased from 7.6 to 4.3 at 3 months, to 5.1 at 6 months and to 5.5 at 12 months.

Regarding the WOMAC score, the improvement after RFA was higher after 3 and 6 months (*p* = 0.012). In the RFA group, it was reduced from 61.2 to 32.9 and 36 at 3 and 6 months, respectively, and in the IAI group from 65.1 to 50 at 3 months and to 53.6 at 6 months.

The NRS was improved in both groups after 3 and 6 months. However, the improvement was more evident after RFA (*p* = 0.019). Specifically, the NRS changed after RFA from 7.1 to 2.6 at 3 and 6 months and after IAI from 7.3 to 4.8 and 5.5 at 3 months and 6 months, respectively.

#### 3.3.3. RFA vs. Steroid IAI

Three studies compared the effectiveness of RFA with that of steroid IAI, including a total number of 139 and 138 patients, respectively [14,31,35]. One of the studies found a statistically significant reduction in VAS score at 3 months following RFA compared to steroid injection (*p* < 0.001). More specifically, the VAS score was improved from 8 to 4 after RFA and from 8 to 5.5 after steroid injection. In the same study, the WOMAC score was reduced from 56.3 to 39.7 and from 47.2 to 42.3 after RFA and steroid injection, respectively. However, no difference between the two groups was identified (*p* = 0.092) [14].

The NRS was evaluated in two studies [31,35]. The improvement at 3 and 6 months was statistically significantly higher following RFA (*p* < 0.05). In detail, the baseline score of 7.3 was reduced to 2.8 at 3 months and to 2.5 at 6 months after RFA. In the steroid group, the NRS was improved from 7.5 to 5.2 and 5.9 at 3 and 6 months, respectively.

As for the OKS, a greater improvement was found after RFA in comparison with steroid injection in one of the three studies (*p* < 0.05) [31]. The score was increased from 16.7 to 34.6 and 35.7 at 3 and 6 months, respectively, after RFA and from 16.9 to 24.6 at 3 months and to 22.4 at 6 months after steroid IAI.

#### 3.3.4. RFA vs. HA IAI

The RFA was compared with the HA injection in three studies including 150 and 148 patients, respectively [30,33,34]. Two studies evaluated the VAS score [33,34], which was improved at 3, 6 and 12 months in both groups but more significantly after RFA (*p* < 0.05). Particularly, following RFA, the VAS score was reduced from 7.9 at baseline to 3.7 at 3 months, to 2.4 at 6 months and to 3.1 at 12 months. After HA IAI, the VAS reduced from 7.9 to 3.5 at 3 months, to 5.1 at 6 months and to 7 at 12 months.

The WOMAC score was calculated in two studies [30,33]. An improvement was found at 3 and 6 months in both groups, which was more significant after RFA (*p* < 0.05). The 74.1 points at baseline were reduced to 22.1 and 33.6 at 3 and 6 months, respectively, after RFA. Furthermore, the 74.1 points at baseline were decreased to 49.7 at 3 months and to 53.6 at 6 months after HA injection.

In one study, the LKS was improved more significantly after RFA compared to HA IAI at 3.6 and 12 months (*p* < 0.05). Following RFA, the 42 points were increased to 68, 83 and 85 at 3, 6 and 12 months, respectively. In HA group, the baseline 41 points were improved to 62 at 3 months, to 54 at 6 months and to 43 at 12 months.

In another study, the NRC, EQ-5D-5L and GPE were statistically significantly improved in favor of RFA when compared to HA injection at 3 and 6 months (*p* < 0.05) [30]. More specifically, the NRS was reduced after RFA from 6.9 to 2.3 and 2.7 at 3 and 6 months, respectively, and after HA from 7 to 4.4 and 5 at 3 and 6 months, respectfully. The EQ-5D-5L was increased from 0.67 to 0.83 at 3 months and to 0.8 at 6 months after RFA. Similarly, there was an increase from 0.66 to 0.75 and 0.72 at 3 and 6 months, respectively, after HA IAI. About the GPE, the reported improvement compared to baseline was found in 79% at 3 months and in 72% at 6 months after RFA and in 51% and 40% at 3 and 6 months, respectively, after HA.

#### 3.3.5. RFA vs. Other Types of IAI

In one of the studies, the RFA was compared with PRP injection, including 100 patients in each group [32]. The RFA group improvement was superior regarding the VAS score at 6 and 12 months, the IKS at 3, 6 and 12 months and patients’ satisfaction according to Likert’s scale at 12 months (*p* < 0.05).

Another study compared the ozone IAI with RFA and found no difference in the VAS score (*p* = 0.2) and OKS (*p* = 0.23) at 12-month follow-up [13].

In a final study, PRP and HA were injected in all included patients but in half of them a RFA was additionally applied [36]. Compared to controls, the RFA group featured a more significant improvement in VAS score, all aspects of AKSS, Physical functioning, Bodily pain, Vitality, General health perceptions and Total aspects of the SF-36 points at 3 months (*p* < 0.05).

#### 3.3.6. Complications

No severe complications were recorded in all studies. Procedural adverse events of RFA were found in six studies and specifically in 36 out of 354 patients (10.2%) [13,30,31,32,35,36]. The most common complication was pain at the sites of ablation (13 cases). Regarding HA injection, one study mentioned that 9 out of the 89 (10.1%) patients had an adverse event and most commonly pain (6 patients—6.8%) during or after the procedure [30]. In two studies, 4 out of 102 (4%) patients treated with steroid IAI experienced some pain during the procedure [31,35]. Pain after IAI of PRP was reported in 2 out of 100 (2%) patients in another study [32]. No adverse event was reported after ozone or combined PRP and steroid IAI in the remaining two studies [13,36]. Based on the above data, the incidence of complications after RFA was similar to HA injections (*p* = 0.523) and higher than steroid or PRP IAI (*p* = 0.023 and *p* = 0.017, respectively).

## 4. Discussion

According to the current systematic review, the RFA seems to be a safe and a more effective technique than IAIs for the treatment of symptomatic knee OA. The pain relief and improved functional outcome after RFA are expected to last for at least 3 months and may continue even 12 months after the procedure. Unfortunately, no data have been provided beyond that time point as all the included studies compared the effect of a single RFA with that of a single IAI. Therefore, no comparison could be made between repeated RFA and IAIs, and the optimal timing of an additional procedure could not be defined. Although one-third of patients may develop some pain at the ablation sites, the overall incidence of adverse events is quite similar compared to IAI options.

The effect of RFA on the alleviation of knee osteoarthritis symptoms has been widely studied and is associated with promising outcomes [37,38]. Li et al. [37], in a meta-analysis of eight randomized control trials, found improvement in pain and WOMAC scores at 6 months after RFA with no serious reported adverse events. Similarly, Ajrawat et al. [39], in a systematic review including 33 studies, correlated RFA with decreased pain and improved function and quality of life for up to 12 months after RFA in patients with symptomatic knee osteoarthritis. On the contrary, in a meta-analysis including five randomized controlled trials comparing RFA with placebo or medication or IAI in patients with knee OA, Fari et al. [40] did not find any statistically significant benefit of RFA despite the overall positive effect of the method. The authors pointed out the heterogeneity of the selected studies and advocated that RFA could be a valid option against severe knee OA, particularly when it was unresponsive to traditional conservative treatment methods. In our review, the RFA was associated with improved function and pain status, which may last for up to 12 months.

Hyaluronic acid has been widely applied for the treatment of knee OA as it may facilitate lubrication, extra cushioning and shock absorption [41]. In a multicenter randomized trial, Chen et al. [30] compared the results of HA and RFA in 82 and 76 knees, respectively, unresponsive to medication-based treatment. After 6 months, RFA was associated with statistically significant pain relief. In addition, and at the same time point, 71% of the RFA users showed more than 50% reduction in pain compared with 38% after intra-articular HA injection (*p* < 0.0001). The RFA was also associated with superior functional outcome and quality of life without showing any serious adverse events. In another randomized trial, Ray et al. [33] randomly assigned 24 patients to receive either percutaneous RFA genicular neurotomy or intra-articular HA injection (12 patients in each group). After 1 week, there was a significant difference in VAS in favor of RFA group. The authors noticed also that there was further decrease in the VAS score after 4 and 12 weeks in RFA group compared to HA group (*p* < 0.0001). Similarly, the knee function, as measured with the WOMAC score, was improved more in RFA group through the 12-week follow-up period. We have also identified that RFA was more effective than HA IAI at 3.6 and 12 months, showing similar complication rate.

Intra-articular corticosteroid injection may reduce knee joint inflammation and swelling and provide pain relief [42]. However, its short-term effect requires repeated injections that could be associated with cartilage damage over an extended exposure time [20,43]. In a prospective, multicenter, randomized trial, Davis et al. [31] compared the clinical safety and effectiveness of RFA with intra-articular steroid injection in patients with OA-related knee pain that failed conservative treatment. A clinically meaningful reduction in knee pain as well as improved joint function were noticed after RFA treatment. All patients were managed on an outpatient basis, and no procedure-related serious adverse events were reported. Sari et al. [14] designed a randomized study in patients with knee OA to investigate if RFA neurotomy of knee genicular nerves was superior to intra-articular steroid injection. Significantly better improvement in pain and knee function was detected after RFA at 1 month and 3 months. Our systematic review revealed a superior clinical outcome at 3- and 6-month follow-up after RFA compared to steroid injections for the treatment of knee OA. Although RFA was associated with more frequent pain during and after the procedure than steroid IAI, no further treatment or hospitalization were deemed necessary and alleviation of symptoms was quickly observed.

Platelet-rich plasma is an autologous blood product that contains a high volume of various growth factors [44]. It can modulate the inflammatory processes and contribute to cartilage preservation or regeneration [45]. Elawamy et al. [32] performed a randomized clinical trial to compare RFA and PRP in terms of knee pain alleviation and function. The latter was estimated with the index of severity of osteoarthritis, which measures three important domains in OA: pain or discomfort, maximum distance walked and activities of daily living. All patients had pain for at least 3 months and advanced knee OA (Grade III–IV according to the K–L Grading Scale). The VAS and index of severity of osteoarthritis were significantly lower in the RFA group after 6 and 12 months. Only 2% in the RFA group experienced pain at the intervention site, which was completely resolved in one week with paracetamol only. In a comparative clinical study, Shen et al. [36] noticed that patients treated with RFA along with PRP and steroid IAI had lower VAS score and higher SF-36 and AKSS scores after 3 months compared to patients treated with PRP and steroid injection alone. Our study showed that the RFA featured better patient-reported outcome measures than the PRP injection treatment at 3-, 6- and 12-month follow-up. However, as both options have been proved effective for the treatment of early to moderate knee OA, a combined therapy with PRP with RFA may further alleviate pain and improve knee function. It seems that integration of both intra-articular and extra-articular treatment modalities can maximize the overall benefit to patients.

This systematic review has certain limitations that are mostly related to the number and type of included studies. Specifically, few studies comparing the RFA with the other conservative treatment modalities have been published so far for the treatment of knee OA. Secondly, various types and concentrations of steroid (betamethasone, methylprednisolone), viscosupplementation and PRP injections were used among the evaluated studies that might led to heterogeneity of the relevant data. Additionally, different RFA protocols and techniques (pulsed RFA, thermal-RFA, cooled RFA) have been introduced for the treatment of chronic knee pain that hamper the extraction of reliable and strong evidence. Finally, due to the multifactorial nature of chronic pain in knee OA, there is lack of consensus regarding which patients may benefit from RFA neurotomy and if the procedure should be re-applied to maximize its effectiveness.

## 5. Conclusions

In patients with symptomatic knee OA, the RFA is associated with more sustained pain relief and better patient satisfaction when compared to IAIs, including HA, steroid and PRP. The effect of RFA can last for at least 3 months and up to 12 months according to the current data. Although no serious adverse events are encountered, the incidence of RFA procedural pain may be higher than injections. Due to the small number of available studies and limited patient population, this result should be interpreted with caution and not be generalized to the entire knee OA cases. Future studies need to concentrate on the role of RFA in knee OA, either alone or combined with other intra-articular treatment options. Furthermore, the issues of optimal injection type, most suitable RFA technique and appropriate timing of a repeated treatment in case of pain recurrence should be better clarified and defined.

## Figures and Tables

**Figure 1 jpm-13-01227-f001:**
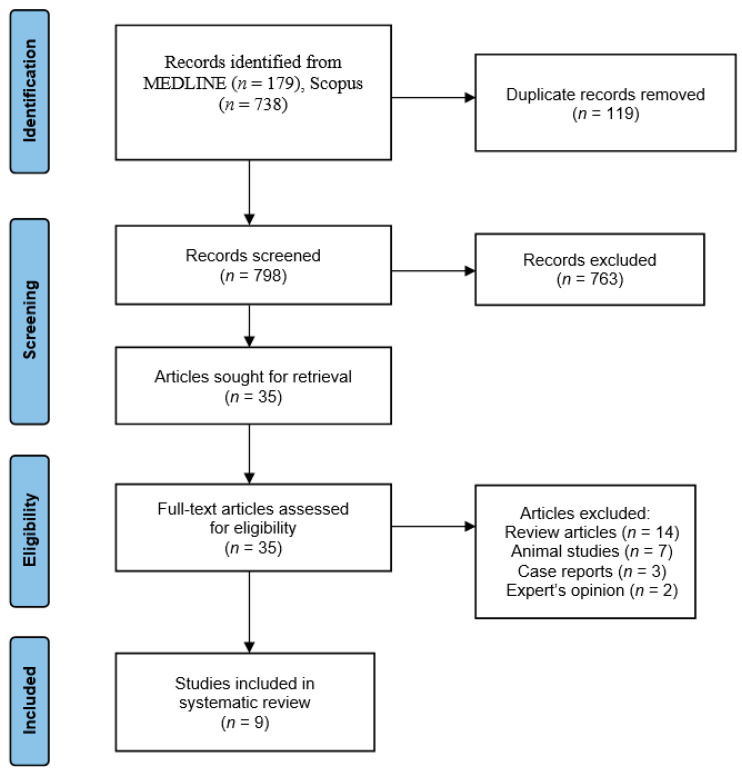
PRISMA flow diagram with research results.

**Table 1 jpm-13-01227-t001:** Summary of risk of bias.

Study	Type	Confounding	Selection of Participants	Classification of Interventions	Randomization Process	Deviations from Intended Interventions	Missing Outcome Data	Measurement of the Outcome	Selection of the Reported Result	Overall
Ray et al. [33]	RCT				3	2	1	1	2	3
Xiao et al. [34]	RCT				1	1	1	1	1	1
Chen et al. [30]	RCT				1	1	1	1	1	1
Davis et al. [31]	RCT				1	1	1	1	1	1
Sari et al. [14]	RCT				1	2	2	1	1	2
Elawamy et al. [32]	RCT				2	1	1	1	1	2
Hashemi et al. [13]	RCT				3	1	2	1	1	3
Shen et al. [36]	Prospective comparative	1	1	1		1	1	1	1	1
Hong et al. [35]	Retrospective comparative	1	1	2		1	1	1	1	2

The Cochrane RoB 2 was used for randomized control trials and the ROBINS-I tool (2026) for other comparative studies. (1: low, 2: some concerns/moderate, 3: high/serious, 4: critical, 5: no information).

**Table 2 jpm-13-01227-t002:** Basic characteristics of the included studies.

Author	Year	Study Type	LoE	Quality *	Risk of Bias **	Level of Certainty ***	K–L	Duration of Symptoms (Months)	Temperature/Duration/Cycles of RFA	NoP RFA Group	Mean Age (Years)	Female (%)	BMI (kg/m^2^)	Intra-Articular Injection—Control Group	NoP Control Group	Follow-Up (Months)
Ray et al. [33]	2018	RCT	1	16	3	1	1, 2, 3	N/A	80 °C/90 s/2	12	51.7	58	N/A	HA	12	3
Xiao et al. [34]	2018	RCT	1	19	1	3	N/A	36.5	60, 70, and 80 °C/90 s/3	49	56.5	76	N/A	HA (sodium hyaluronate; Chemical Industries Ltd., Shizuoka, Japan)	47	12
Chen et al. [30]	2020	RCT	1	21	1	4	N/A	90	60 °C/150 s/1	89	63.3	58	32.2	HA (Synvisc-One (hylan G-F 20); Sanofi, Paris, France)	87	6
Shen et al. ^¥^ [36]	2017	Prospective comparative	2	21	1	3	N/A	60	70 °C/120 s/1	27	62.2	74.1	N/A	HA (sodium hyaluronate) + platelet-rich plasma	27	3
Davis et al. [31]	2018	RCT	1	21	1	3	2, 3, 4	128	60 °C/150 s/1	76	63	65.8	30.6	Steroid (40 mg methylprednisolone acetate or triamcinolone topical or betamethasone)	75	6
Sari et al. [14]	2018	RCT	1	19	2	2	2, 3, 4	60	80 °C/90 s/1	37	64	81.1	23.5	Steroid (1 mL betamethasone + 2.5 mL bupivacaine + 2.5 mg morphine)	36	3
Hong et al. [35]	2020	Retrospective comparative	3	20	2	3	2, 3	32.5	70 °C/180 s/1	26	59.5	61.5	24.6	Steroid (2 mg betamethasone sodium phosphate + 5 mg betamethasone dipropionate)	27	6
Elawamy et al. [32]	2021	RCT	1	21	2	2	3, 4	94.8	42 °C/120 s/3	100	47.8	50	N/A	Platelet-rich plasma	100	12
Hashemi et al. [13]	2016	RCT	1	18	3	1	2, 3	N/A	70 °C/90 s/1	36	68.3	77.8	26.1	Ozone (10 cc O_2_–O_3_ mixture 40 μg/mL) (+ periarticular injection (5 cc O_2_–O_3_ mixture 10 μg/mL))	36	12

Abbreviations: HA: hyaluronic acid, K–L: Kellgren–Lawrence classification, LoE: level of evidence, NoP: number of patients, N/A: not applicable, RCT: randomized control trial, RFA: radiofrequency ablation. ^¥^ in study group, RFA was performed along with HA + platelet-rich plasma. * The Critical Appraisal Skills Programme (CASP) was used for randomized control trials, and the revised methodological index for non-randomized studies (MINORS) was used for other comparative studies. ** The Cochrane RoB 2 was used for randomized control trials and the ROBINS-I tool (2026) for other comparative studies (1: low, 2: some concerns/moderate, 3: high/serious). *** The level of certainty was measured using the GRADE system (1: very low, 2: low, 3: moderate, 4: high).

**Table 3 jpm-13-01227-t003:** Outcomes of RFA and comparison with IAI modalities.

Author	Year	Control	Scores	Time of Evaluation	Outcome
Ray et al. [33]	2018	HA	VAS, WOMAC	Bsl, 1 w, 1 m, 3 m	RFA Improved * VAS and WOMAC at 1 w–1 m–3 m (*p* < 0.001) RFA vs. Control RFA: Better VAS at 1 w (*p* < 0.004)–1 m–3 m and WOMAC at 1 w–1 m–3 m (*p* < 0.001)
Xiao et al. [34]	2018	HA	VAS, LKS	Bsl, 3 d, 3 m, 6 m, 9 m, 12 m	RFA Improved * VAS and LKS at 3 d–3 m–6 m–9 m–12 m (*p* < 0.05) RFA vs. Control RFA: Better VAS and LKS at 3 d–3 m–6 m–9 m–12 m (*p* < 0.05)
Chen et al. [30]	2020	HA	NRS, WOMAC, GPE, EQ-5D-5L	Bsl, 1 m, 3 m, 6 m	RFA Improved * all aspects of WOMAC at 1 m–3 m–6 m, EQ-5D-5L at 6 m (*p* < 0.05) RFA vs. Control RFA: Better WOMAC total-pain-physical function at 1 m–3 m–6 m, WOMAC stiffness at 3 m–6 m, improvement in GPE at 1 m–3 m–6 m, EQ-5D-5L 1 m–3 m–6 m (*p* < 0.05)
Shen et al. ^¥^ [36]	2017	HA + platelet-rich plasma	VAS, AKSS, SF-36	Bsl, post, 3 m	RFA Improved * VAS at post-3 m, all aspects of AKSS at post-3 m, Physical functioning-Bodily pain-General health perceptions-Vitality-Social role functioning-Total points of SF-36 at post-3 m (*p* < 0.05) RFA vs. Control RFA: Better VAS at post-3 m, all aspects of AKSS at post-3 m, Physical functioning-Bodily pain-Vitality-Total points of SF-36 at post-3 m, General health perceptions of SF-36 at 3 m (*p* < 0.05)
Davis et al. [31]	2018	Steroid	NRS, OKS, GPE	Bsl, 1 m, 3 m, 6 m	RFA Improved * NRS and OKS at 1 m–3 m–6 m (*p* < 0.05) RFA vs. Control RFA: Better NRS and OKS at 1 m–3 m–6 m, improvement in GPE at 3 m–6 m (*p* < 0.05)
Sari et al. [36]	2018	Steroid	VAS, WOMAC	Bsl, 1 m, 3 m	RFA Improved * VAS at 1 m–3 m, WOMAC at 1 m–3 m (*p* < 0.001) RFA vs. Control RFA: Better VAS at 1 m–3 m, WOMAC total at 1 m, WOMAC stiffness at 3 m, WOMAC function at 1 m (*p* < 0.001)
Hong et al. [35]	2020	Steroid	NRS, OKS, GPE	Bsl, 1 w, 1 m, 3 m, 6 m	RFA Improved * NRS and OKS at 1 w–1 m–3 m–6 m (*p* < 0.05) RFA vs. Control RFA: Better NRS at 1 m–3 m–6 m, OKS at 3 m–6 m, improvement in GPE at 3 m–6 m (*p* < 0.05)
Elawamy et al. [32]	2021	Platelet-rich plasma	VAS, ISK, Likert’s scale of satisfaction	Bsl, 1 w, 3 m, 6 m, 12 m	RFA Improved VAS and ISK at 1 w–3 m–6 m–12 m (*p* < 0.05) RFA vs. Control RFA: Better VAS at 6 m–12 m, ISK at 1 w–3 m–6 m–12 m, satisfaction-Likert’s scale at 12 m (*p* < 0.05)
Hashemi et al. [13]	2016	Ozone	VAS, OKS	Bsl, 12 m	RFA Improved * VAS at 12 m, OKS at 12 m (*p* < 0.05) RFA vs. Control No difference in VAS 12 m, OKS 12 m

Abbreviations: AKSS: American Knee Society Score, Bsl: baseline, d: day, EQ-5D-5L: EuroQol 5 Dimension 5 Level, GPE: Global Perceived Effect, HA: Hyaluronic acid, IKS: Index of Severity for Osteoarthritis of the Knee, LKS: Lysholm Knee score, m: month, NRS: Numeric Rating Scale, OKS: Oxford Knee Scale, post: after intervention, RFA: radiofrequency ablation, SF-36: 36-Item Short Form Survey, VAS: Visual Analogue Scale, w: week, WOMAC: Western Ontario and McMaster University Osteoarthritis Index. ^¥^ in study group, RFA was performed along with HA + platelet-rich plasma. * compared to the baseline.

## Data Availability

Not applicable.

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
