# Peer review of "Is Radiofrequency Ablation Superior to Intra-Articular Injections for the Treatment of Symptomatic Knee Osteoarthritis?—A Systematic Review"

_jpm, 2023, doi:10.3390/jpm13081227_

Round 1
Reviewer 1 Report
The authors of this article have reviewed the literature on radiofrequency ablation (RFA) on knee osteoarthritis. This review has been conducted according to the PRISMA guidelines and is included in a figure within the manuscript. The conclusion from the review is that RFA has greater effectiveness in treating symptomatic OA compared to intra-articular injections.
The authors need to consider the following in their manuscript,
1. RFA fundamentally reduces pain in OA patients by reducing nerve pain. Thus, is the treatment more of a short-term fix that leads to further degeneration in the long term ?
2. What is the equivalence of repeated injection with respect to RFA ? How many injections were used in the described studies ?
3. It is stated that there are two different versions of the OKS score system. How accurate are these measurements and are they comparable to each other ? A statement on this part should be made.
4. There are comparisons between RFA and different intra-articular injection types. How comparable are the intra-injection data between the reviewed manuscripts. Are they consistent or are there differing results reflective of different injection regimes or patient cohort ? A comment on this part should also be made.
A minor English correction is required.
Author Response
Dear Editor,
We would like to thank you for accepting to reconsider our manuscript titled: “Is Radiofrequency Ablation Superior to Intra-articular Injections for the Treatment of Symptomatic Knee Osteoarthritis? A Systematic Review.” for publication in the Journal of Personalized Medicine for the special issue “Management of Osteoarthritis Pain”.
We would also like to thank the reviewers for their insightful comments. All raised points have been addressed and the manuscript has been revised according to their suggestions. All text changes in the manuscript have been highlighted. For reviewing purposes, the comments have been addressed one by one.
In more detail:
Reviewer #1
Comment: “RFA fundamentally reduces pain in OA patients by reducing nerve pain. Thus, is the treatment more of a short-term fix that leads to further degeneration in the long term ?”
Reply: Thank you for your comment. RFA causes separation in myelinated axons of the target nerves, but any long-term nerve degeneration has not been proven yet. This issue is presented in detail in the “Introduction” (lines 50-53).
Comment: “What is the equivalence of repeated injection with respect to RFA ? How many injections were used in the described studies?”
Reply: Thank you for the comment. All the included studies compared the effect of a single RFA with that of a single intra-articular injection. Therefore, no comparison or correlation could be made for repeated RFA and intra-articular injection procedures. A relevant comment has been inserted in the 1st paragraph of Discussion section (lines 281-284)
Comment: “It is stated that there are two different versions of the OKS score system. How accurate are these measurements and are they comparable to each other? A statement on this part should be made.”
Reply: Thank you for the comment. There are two versions of OKS score system. According to the old version, a lower score reflects a better outcome. On the contrary, and in the latest version of OKS, a higher score indicates better performance and less disability. This information has been added in the manuscript (lines 181-183)
Comment: “There are comparisons between RFA and different intra-articular injection types. How comparable are the intra-injection data between the reviewed manuscripts. Are they consistent or are there differing results reflective of different injection regimes or patient cohort ? A comment on this part should also be made.”
Reply: Thank you for the comment. We totally agree that there were used different types of steroid (betamethasone, methylprednisolone), viscosupplementation and PRP injections in the studies. The heterogeneity of these data is one of the limitations of the present systematic review. A relevant comment has been added in the Discussion section (lines 355-357).
Reviewer 2 Report
Dear authors, thank you for allowing me to participate as peer reviewer of your manuscript. I consider that in general terms the methodology that you use for conducting the SR was fine; however it deserves some improvements in the methods, results analysis and discussion part related to the analysis and description of the findings. Hereafter I describe below the concepts that should be added to the different parts of the manuscript to complete the content of the review:
Jpm-2489365
Is Radiofrequency Ablation Superior to Intra-articular Injections for the Treatment of Symptomatic Knee Osteoarthritis? A Systematic Review.
Title and introduction
Please avoid to use the same words of the title as key words.
Use past tense in the lines 56.
Please provide registration information of the review, including register name and registration number.
Methods
Please add to the Prisma flow chart the reasons for the exclusion of the articles in the elegibility and screening step. ( you could follow the suggested flow diagram template in M.J. Page et al./Journal of Clinical Epidemiology 134 (2021) 178–189.
In materials and methods point 2.2 should be called “Search strategy”. Please add the full search strategies for all data bases, including the filters and limits used. Please include it as asupplement.
Please describe any methods used to assess risk of bias due to missing results in a synthesis.
Please define in data analysis part the continuous and categorical variables that were considered for the analysis, include the methods you used to explore possible causes of heterogeneity among study results.
Please describe any methods used to assess certainty ( or confidence) in the body of evidence for the outcome and point out the level of certainty.
Results
Please add the risk of bias of each study within the table 1.
Present the assessment of the risk of bias of each study.
Please present the causes of possible heterogeneity among studies.
Present the results of assessments of certainty for each outcome assessed.
Discussion
Please discuss any limitations of the evidence included in the review.
Please discuss implications of the results for practice, policy and future research.
Author Response
Dear Editor,
We would like to thank you for accepting to reconsider our manuscript titled: “Is Radiofrequency Ablation Superior to Intra-articular Injections for the Treatment of Symptomatic Knee Osteoarthritis? A Systematic Review.” for publication in the Journal of Personalized Medicine for the special issue “Management of Osteoarthritis Pain”.
We would also like to thank the reviewers for their insightful comments. All raised points have been addressed and the manuscript has been revised according to their suggestions. All text changes in the manuscript have been highlighted. For reviewing purposes, the comments have been addressed one by one.
Reviewer # 2
Comment: “Please avoid to use the same words of the title as key words.”
Reply: Thank you for the comment. We have changed the keywords in order to differ from those in Title.
Comment: “Use past tense in the lines 56.”
Reply: We changed the relevant sentence according to your suggestion (line 66).
Comment: “Please provide registration information of the review, including register name and registration number.”
Reply: Thank you for the comment. According to your instructions, the study was registered in PROSPERO database.
Comment: “Please add to the Prisma flow chart the reasons for the exclusion of the articles in the eligibility and screening step. (you could follow the suggested flow diagram template in M.J. Page et al./Journal of Clinical Epidemiology 134 (2021) 178–189.”)
Reply: The PRISMA flow diagram was modified and re-written according to your instructions. (line 140)
Comment: “In materials and methods point 2.2 should be called “Search Strategy”.
Reply: Thank you for the comment. We changed the 2.2 heading to “Search Strategy”. (line 74)
Comment: “Please add the full search strategies for all databases, including the filters and limits used. Please include it as a supplement.”
Reply: Thank you for the comment. The search strategy and the filters applied are described and presented in sections 2.2 and 2.3. (lines 74-90)
Comment: “Please describe any methods used to assess risk of bias due to missing results in a synthesis.”
Reply: Following your comment, the risk of bias was assessed using the Cochrane RoB 2 tool for randomized control trials and the ROBINS-I tool (2026) for non‐randomized intervention studies.
Comment: “Please define in data analysis part the continuous and categorical variables that were considered for the analysis, include the methods you used to explore possible causes of heterogeneity among study results.”
Reply: Thank you for the comment. The continuous and categorical variables that were considered for data analysis are presented in “Data Synthesis and Analysis” section. In addition, the heterogeneity among studies was assessed using the Cochrane’s Q test. (line 122)
Comment: “Please describe any methods used to assess certainty (or confidence) in the body of evidence for the outcome and point out the level of certainty.”
Reply: Following your comment, the GRADE system was used to assess the certainty. Moreover, the level of certainty was added in table 2. (line 164)
Comment: “Please add the risk of bias of each study within the table 1.”
Reply: Thank you for the comment. We added the risk of bias of each study in Table 2. (line 164)
Comment: “Present the assessment of the risk of bias of each study.”
Reply: According to your instruction, we added Table 1 with the assessment of the risk of bias of every study 162.
Comment: “Please present the causes of possible heterogeneity among studies.”
Reply: Thank you for the comment. Any heterogeneity among studies was mainly due to various types and concentrations of steroid (betamethasone, methylprednisolone), viscosupplementation and PRP injections as well as RFA techniques that used in the included studies. However, and according to Cochrane’s Q test, no significant heterogeneity of the included studies was found. This issue is presented in limitations paragraph of Discussion section. (lines 352-362)
Comment: “Present the results of assessments of certainty for each outcome assessed.”
Reply: Following your comment, the results of the assessment of certainty are presented in Table 2.
Comment: “Please discuss any limitations of the evidence included in the review.”
Reply: Thank you for the comment. The limitations of the current study are discussed in the final paragraph of the Discussion. (lines 352-362)
Comment: “Please discuss implications of the results for practice, policy and future research.”
Reply: Following your comment, the issues of clinical application and practice as well as future research directions are presented in the last two paragraphs of Discussion and Conclusion sections of the manuscript. (lines 350, 368-374)
Reviewer 3 Report
Dear Sirs,
your paper is interesting but many concerns have to be better addressed.
Introduction seems poor; here you should explain in which way your review is different from other recent review on the same issue, such as the following: Wu L, Li Y, Si H, et al. Radiofrequency Ablation in Cooled Monopolar or Conventional Bipolar Modality Yields More Beneficial Short-Term Clinical Outcomes Versus Other Treatments for Knee Osteoarthritis: A Systematic Review and Network Meta-Analysis of Randomized Controlled Trials. Arthroscopy. 2022;38(7):2287-2302. doi:10.1016/j.arthro.2022.01.048
Materials and methods is lacking. First of all, a PROSPERO registration is missing. Then, it is not clear how you selected the studies. Did you select only randomized trial? How did You excluded 763 articles in the screening phase? Which was the primary outcome? If it was a functional scale, which one? How did you perform a statistical analysis using different outcomes? You have to justify all these aspects.
Discussion should be integrated. This statement is not supported by the current literature: "According to the current systematic review, the RFA seems to be a safe and a more 250 effective technique than IAIs for the treatment of symptomatic knee OA". Moreover, a deep comparison with the available literature about RF effectiveness is needed. To do that and to reinforce your hypothesis, I suggest the following reference: Farì G, de Sire A, Fallea C, et al. Efficacy of Radiofrequency as Therapy and Diagnostic Support in the Management of Musculoskeletal Pain: A Systematic Review and Meta-Analysis. Diagnostics (Basel). 2022;12(3):600. Published 2022 Feb 26. doi:10.3390/diagnostics12030600
For the same reasons mentioned above, your conclusion about RF superiority in comparison with injections seem too ambitious.
Best regards and good lucke
Author Response
Dear Editor,
We would like to thank you for accepting to reconsider our manuscript titled: “Is Radiofrequency Ablation Superior to Intra-articular Injections for the Treatment of Symptomatic Knee Osteoarthritis? A Systematic Review.” for publication in the Journal of Personalized Medicine for the special issue “Management of Osteoarthritis Pain”.
We would also like to thank the reviewers for their insightful comments. All raised points have been addressed and the manuscript has been revised according to their suggestions. All text changes in the manuscript have been highlighted. For reviewing purposes, the comments have been addressed one by one.
In more detail:
Reviewer # 3
Comment: “Introduction seems poor; here you should explain in which way your review is different from other recent review on the same issue, such as the following: Wu L, Li Y, Si H, et al. Radiofrequency Ablation in Cooled Monopolar or Conventional Bipolar Modality Yields More Beneficial Short-Term Clinical Outcomes Versus Other Treatments for Knee Osteoarthritis: A Systematic Review and Network Meta-Analysis of Randomized Controlled Trials. Arthroscopy. 2022;38(7):2287-2302. doi:10.1016/j.arthro.2022.01.048”
Reply: Thank you for your comment. In contrast to previous systematic reviews, which compared the radiofrequency ablation (RFA) with different types of conservative treatment of knee osteoarthritis, the current one directly compares the RFA with all types of intra-articular injections (IAI). This issue is better presented in Introduction section. (lines 61-65)
Comment: “Materials and methods is lacking. First of all, a PROSPERO registration is missing. Then, it is not clear how you selected the studies. Did you select only randomized trial? How did You excluded 763 articles in the screening phase? Which was the primary outcome? If it was a functional scale, which one? How did you perform a statistical analysis using different outcomes? You have to justify all these aspects.”
Reply: Thank you for the comments. 1) Following your instructions, the study was registered in PROSPERO database 2) The inclusion criteria and search strategy are presented in 2.2 and 2.3 sections (lines 82-85). 3) The process of excluded articles is described in 3.1 section. (lines 127-134). 4) The primary outcome of the study was the comparison of the effect of RFA and IAI on knee pain, using the Visual Analogue Scale (VAS) and the Numeric Rating Scale (NRS) scores (2.4 section, lines 97-99). 5) The statistical analysis was performed independently for each examined parameter. (lines 118-121)
Comment: “Discussion should be integrated. This statement is not supported by the current literature: "According to the current systematic review, the RFA seems to be a safe and a more 250 effective technique than IAIs for the treatment of symptomatic knee OA". Moreover, a deep comparison with the available literature about RF effectiveness is needed. To do that and to reinforce your hypothesis, I suggest the following reference: Farì G, de Sire A, Fallea C, et al. Efficacy of Radiofrequency as Therapy and Diagnostic Support in the Management of Musculoskeletal Pain: A Systematic Review and Meta-Analysis. Diagnostics (Basel). 2022;12(3):600. Published 2022 Feb 26. doi:10.3390/diagnostics12030600”
Reply: Thank you for your comment and suggestion. The study of Fari et al is thoroughly presented in Discussion section (lines). Their systematic review included 5 RCT knee studies comparing the RFA with other treatments and only in 3 of them an injection was applied in the control group. We agree that despite our result of the effect of RFA in knee OA and due to the small number of available studies and limited patient population, this result should be interpreted with caution and not be generalized to the entire knee OA population. This statement has been added in Abstract and Conclusions sections. (lines 23-25, 368-374)
Comment: “For the same reasons mentioned above, your conclusion about RF superiority in comparison with injections seem too ambitious.”
Reply; Thank you again for the comment. According to the previous response, we clearly point out in the manuscript that despite the results of the statistical analysis, the potential superiority of the RFA over the IAIs should be interpreted with caution due to the limited available data, the small number of patients and the heterogeneity of the studies. (lines 23-25, 368-370)
Round 2
Reviewer 1 Report
The authors have addressed my comments appropriately and this manuscript can be considered for publication.
A minor English correction and review is required by the authors.
Reviewer 2 Report
Dear authors the manuscript has improved sufficiently to warrant publication in JPM.
Reviewer 3 Report
Dear Sirs,
thank you for the effort to improve the quality of your paper according to my suggestions.
No further corrections are needed.
Regards